# The Role of the Adenoids in Pediatric Chronic Rhinosinusitis

**DOI:** 10.3390/medsci7020035

**Published:** 2019-02-25

**Authors:** Ryan Belcher, Frank Virgin

**Affiliations:** Department of Otolaryngology, Vanderbilt University Medical Center, Vanderbilt University, Nashville, TN 37235, USA; ryan.belcher@vumc.org

**Keywords:** adenoids, adenoiditis, pediatric, chronic rhinosinusitis, adenoidectomy, sinusitis, biofilm

## Abstract

There are several mechanisms by which the adenoids contribute to pediatric chronic rhinosinusitis (PCRS), particularly with children aged 12 years and younger. Understanding the role that the adenoids play in PCRS is crucial when attempting to treat these patients. A literature review was performed to address this problem and provide information surrounding this topic. This review will provide a better understanding of how adenoids contribute to PCRS, and also of the medical and surgical treatment options.

## 1. Introduction

Pediatric chronic rhinosinusitis (PCRS) is a condition commonly encountered in otolaryngology practice. Pediatric patients acquire a large burden of upper respiratory tract infections (URTI), with 5–13% of these URTIs progressing to acute bacterial sinusitis, and a proportion of those progressing to PCRS [1]. The disease is diagnosed in 2.1% of children in ambulatory healthcare visits per year in the United States and is known to have a significant impact on health-related quality of life [2]. Due to the increased awareness of the disease’s prevalence, over the last eight years multiple professional societies, which include the American Rhinologic Society (ARS) [3], the American Academy of Otolaryngology—Head and Neck Surgery (AAO-HNS) [1], the European Rhinologic Society, and the European Academy of Allergy and Clinical Immunology [4], as well as the Canadian Society of Otolaryngology—Head and Neck Surgery [5], have written clinical consensus statements addressing optimal management of these patients.

There are many factors that contribute to the development of PCRS, including the adenoids, impairment in mucociliary clearance (e.g., primary ciliary dyskinesia, and cystic fibrosis), and anatomic abnormalities of the sinuses, among many others. A major difference between the pathophysiology of the disease process for children compared to adults is the role of the adenoid pad. Adenoids have been shown to have a significant impact on the development of PCRS in children aged 12 years and younger [1,6]. 

Medical management is considered the first line of therapy in the treatment of PCRS, with surgical intervention reserved for patients who fail to improve with these conservative measures [1]. Due to the understanding of the role the adenoid pad plays in PCRS, adenoidectomy with or without maxillary irrigation is the most commonly performed surgical intervention reported among members of ARS and the American Society of Pediatric Otolaryngology (ASPO) [7,8]. However, evidence has not shown a role for adenoidectomy in pediatric patients 13 years or older [1,9]. 

Several studies have been performed to evaluate the relationship between the adenoid pad and paranasal sinus in pediatric patients with chronic rhinosinusitis (CRS), and this review looks to summarize these studies, specifically looking at disease presentation, adenoiditis, obstruction, biofilm formation, colonization, immune function, and medical and surgical management. 

## 2. Disease Presentation/Diagnosis

Pediatric chronic rhinosinusitis is defined as at least 90 continuous days of symptoms of purulent rhinorrhea, nasal obstruction, facial pressure/pain, or a cough with corresponding endoscopic and/or computed tomography (CT) findings in a patient who is 18 years of age or younger [1]. Thus obtaining a thorough and complete history and physical is important for establishing a diagnosis. It is important to note that age is a distinguishing factor in the diagnosis of PCRS in that allergic rhinitis is a more prominent factor in older children, whereas adenoid disease (independent of adenoid size) is a more important contributing factor in younger children [1].

As the PCRS definition states, CT is the gold standard for imaging when establishing a PCRS diagnosis or preparing for sinus surgery, particularly a non-contrasted CT with axial, coronal, and sagittal views. The sensitivity and specificity of plain radiographs, such as a lateral soft tissue neck X-ray evaluating the adenoids, are limited in evaluating the patient’s need for adenoidectomy [10,11]. According to the AAO-HNS consensus on appropriate use of CT imaging, it is recommended in patients with PCRS when medical management and/or adenoidectomy have failed to control symptoms [11]. Practice patterns differentiate between otolaryngologists on whether to obtain CT imaging before adenoidectomy with 82% of surveyed ASPO members elected to perform adenoidectomy prior to obtaining a CT scan compared to 40% of surveyed ARS members (*p* < 0.001) [7,8]. Bhattacharyya et al., in 2004, showed that a CT with a Lund‒Mackay score of ≥5 had good sensitivity and specificity in establishing a diagnosis of PCRS in children not responsive to medical treatment [12]. 

Nasal endoscopy is another option in the armamentarium of otolaryngologists in the evaluation of PCRS. While nasal endoscopy can be useful in the diagnosis of PCRS, it can also be used to diagnose adenoiditis and adenoid hyperplasia. In a survey of otolaryngologists, 48% reported that they always or almost always use nasal endoscopy to establish a diagnosis of PCRS, with 21% reporting usually, and 26% sometimes [8].

## 3. Pathophysiology of Adenoid Contribution to Pediatric Chronic Rhinosinusitis

Adenoid tissue is implicated in contributing to PCRS by several different mechanisms, which include serving as a bacterial reservoir and causing posterior nasal obstruction [13]. Both of these are thought to be factors in causing impaired mucociliary clearance of the sinus cavities. Similar to the sinus mucosa, the adenoids are lined by a layer of ciliated epithelium that can undergo metaplastic change and loss of cilia as a consequence of recurrent or chronic inflammation [13,14]. Posterior nasal obstruction can cause mucous retention in the sinus cavity and in the adenoid pad, which in turn can cause microbial colonization and subsequent inflammation of the mucosa. When an adenoidectomy is performed in this context it relieves the posterior nasal obstruction and removes a significant bacterial reservoir, allowing for better clearance of nasal secretions. Decreased nasal mucosal inflammation can result in improved mucociliary clearance and lead to less sinus ostial obstruction from mucosal edema and better sinus ventilation and drainage [13]. In an attempt to prove this concept, Arnaoutakis et al. used Andersen’s saccharine test in 10 patients with adenoid hypertrophy, chronic adenoiditis, or PCRS to measure the nasal mucociliary clearance time (MCT) and mucociliary velocity (MCV) before and after adenoidectomy. They found that both MCT and MCV improved postoperatively among the group, which was considered to be clinically relevant. However, the small population size precluded testing for statistical significance [13]. 

Bacterial biofilms have been shown to cover tonsillar, adenoid, and sinus mucosa. In comparing PCRS patients vs. obstructive sleep apnea (control) patients receiving adenoidectomy, Zuliani et al. demonstrated identifiable biofilms covering almost the entire mucosal surface on all adenoid specimens from PCRS patients and no biofilms from the obstructive sleep apnea patients [15]. Biofilms can be problematic due to their decreased metabolic activity and expression and transmission of resistance genes. These characteristics may lead to decreased or incomplete penetration of antimicrobials as well as unique antimicrobial resistance patterns [15]. This can also allow sinus microbials to persist in the nasopharynx in PCRS, resulting in minimal to no improvement after frequent antibiotic courses administered. Not surprisingly, biofilms themselves have been shown to cause a state of chronic inflammation to the surrounding tissue that may not even be involved in the microbial infections [15]. The most common pathogenic organisms identified in the adenoids include *Staphylococcus aureus*, *Streptococcus pneumoniae*, *Haemophilus influenzae*, and group A *streptococci* [16]. These same bacteria are similar to the common organisms found in acute and chronic sinusitis in children [17]. The presence of biofilms on adenoid tissue in PCRS is another example that supports the role of an adenoidectomy in that it mechanically removes a potential nidus for re-infection of the sinuses.

The adenoids are covered in respiratory epithelium and are therefore considered in part a secretory immunological organ that provides local secretory immunoglobulin A (IgA) that contributes directly to regional surface protection. Immunoglobulin A is an important immunoglobulin in the upper respiratory tract as it binds to bacteria and suppresses colonization. When compared to controls, Eun et al. found that adenoid tissue in patients with otitis media (OM) and PCRS had a significantly lower amount of IgA (*p* = 0.016 and *p* = 0.004, respectively). They postulate that the increased susceptibility to infection in these patients was likely caused by the reduction in IgA [18]. Further research in the immunological side of PCRS and adenoids is needed to elucidate whether the decreased IgA is caused by concomitant inflammation or if these patients are innately deficient in IgA in their upper respiratory tract, making them more susceptible to chronic inflammation and subsequent PCRS. 

Even though posterior nasal obstruction is thought to contribute to PCRS, no studies have been able to correlate the size of the adenoid pad with the presence of sinonasal symptoms in PCRS. No association has been found, whether the studies have looked at radiographic evidence of nasopharyngeal obstruction by adenoid hypertrophy [19], at the volume [20], or at the weight of adenoid tissue removed [21]. Another potential causative agent that has been investigated for adenoiditis and/or PCRS is *Helicobacter pylori*. The majority of studies evaluating this relationship have been performed in the adult population; however, there are an increasing number of studies being performed in the pediatric population as more non-invasive techniques for detecting *H. pylori* have been developed. In the pediatric studies, regardless of technique, no studies have consistently been able to identify *H. pylori* or structures compatible with the microorganism with most studies finding no evidence of bacteria in their samples [22]. Recently, Grateron Cedeno et al. failed to detect the presence of *H. pylori* in adenoid tissue or maxillary sinus in PCRS patients despite using high-sensitivity and -specificity diagnostic techniques. In their conclusion they emphasize an unlikely role of the microorganism in PCRS without nasal polyps and adenoidal hypertrophy and/or chronic adenoiditis etiology [22].

## 4. Treatment

Whether the adenoids are implicated or not, medical management is considered the first line of therapy in the treatment of PCRS. The duration and combination of medications that constitute “maximal medical therapy” is still under debate. Studies have shown that a topical nasal steroid spray and daily, topical nasal irrigations are beneficial medical therapies [1]. Once-daily nasal saline irrigations have been shown to improve quality of life and Lund‒Mackay scores after just six weeks in PCRS [23]. The use of antimicrobials in the treatment of PCRS is known to be widespread, with no agreed-upon optimal duration. The AAO-HNS consensus statement reports that 20 days of antibiotic therapy may produce a superior response in PCRS patients compared to 10 days of antibiotics. However, the panel failed to reach consensus on the appropriate antibiotic duration in PCRS, but stated that it should be a minimum of 10 consecutive days [1]. It was also agreed upon that culture-directed antibiotics may improve outcomes when patients have not responded to empirical antibiotic treatment [1]. 

A survey of ASPO members evaluated their members’ preferences when it comes to maximal medical therapy. One hundred fifteen members responded to the survey. Within this response group, the most common medications used within their “maximal medical” therapy were nasal steroid sprays (96%), saline irrigations (93%), and oral antibiotics (91%) [7]. Less commonly included medications in their regimens were oral steroids (43%), oral antihistamines (38%), anti-leukotrienes (36%), anti-reflux medications (26%), nasal antihistamine sprays (20%), nasal steroid irrigations (19%), nebulized antibiotics or steroids (7%), and intravenous antibiotics (3%) [7]. When it came to utilizing antibiotics, 65% treat for 15–21 days, 24% treat for >21 days, and 11% for <14 days [7]. Of note, this ASPO survey was completed two years after the AAO-HNS PCRS clinical consensus statement was published.

## 5. Surgical Treatment

Surgical intervention is reserved for patients with PCRS that have failed “maximal medical therapy.” There are several options for surgery in these patients including adenoidectomy and endoscopic sinus surgery (ESS), which have an age- and anatomy-dependent differentiation. Given the role that the adenoids have been shown to play in the etiology of PCRS for children 12 years and younger, adenoidectomy should be considered as a first-line surgical option [1]. It is a simple, well-tolerated procedure and a meta-analysis evaluating the efficacy of adenoidectomy alone in the PCRS population demonstrated a success rate of approximately 70%, in which patients had improved sinusitis symptoms after intervention [24]. Although the tonsils are a part of Waldeyer’s ring and have similar bacteriology, tonsillectomy is considered an ineffective treatment for PCRS [1]. Ramadan et al. showed that children with chronic adenoiditis, which has similar symptoms to PCRS, who had a CT scan with a Lund‒Mackay score of ≤5 and received an adenoidectomy had a higher success rate than children with chronic adenoiditis and CRS (65% versus 43%; *p* = 0.0017) [25]. This study also showed that asthmatic children with CRS had a very poor response rate to adenoidectomy alone when compared with asthmatic children with CA (28% versus 53%; *p* = 0.022) [25]. 

Adjunctive procedures can also be performed at the same time as the adenoidectomy; most commonly, these are maxillary sinus irrigations or balloon sinuplasty. Ramadan et al. showed that adenoidectomy with the addition of maxillary sinus irrigation in PCRS resulted in improved one-year outcomes (87.5%) vs. adenoidectomy alone (60.7%) [26]. Since the U.S. Food and Drug Administration approved balloon catheter sinuplasty in 2006 it has emerged as a potential treatment option for PCRS. Several studies have evaluated the role of balloon sinuplasty in PCRS. Ramadan et al. [27] performed balloon sinuplasty alone versus adenoidectomy in a group of PCRS patients. This study demonstrated superior improvement in symptoms at one year in the balloon group (80%) vs. adenoidectomy group (52.6%) [28]. A more recent randomized, blinded study that evaluated the impact of adding balloon catheter dilation to adenoidectomy with maxillary irrigation did not demonstrate any improved benefit at one year [6]. 

In patients in which PCRS is persistent despite adenoidectomy, the otolaryngologist can then consider performing an ESS. There is a lack of convincing evidence at this time that ESS causes a clinically significant impairment of facial growth in children with CRS, so may be appropriate at any age [1]. In order to optimize the outcomes of all treatments or interventions, appropriately treating concomitant issues such as asthma has been shown to improve the outcomes of PCRS. Accordingly, there is evidence that clinical control of PCRS is important in aiding in the control of asthma [29]. Other factors or underlying diagnoses that may lead to persistent or recalcitrant PCRS despite interventions should be considered and investigated further, such as cystic fibrosis, allergies, primary ciliary dyskinesia, and immunodeficiencies. 

## 6. Conclusions

The adenoid pad plays a key role in the etiology of PCRS through several mechanisms, particularly for children aged 12 and younger. Should these PCRS patients fail maximal medical therapy, adenoidectomy +/− adjunctive treatments should be considered first-line surgical intervention.

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
