# Peer review of "The Role of the Adenoids in Pediatric Chronic Rhinosinusitis"

_medsci, 2019, doi:10.3390/medsci7020035_

Reviewer 1 Report

There is no information about;

-Which bacteria are shown in PCRS and adenoid tissues?

-What is maximal medical therapy for PCRS?

-Which antibiotics are recommended by professional society guidelines for PCRS?

Author Response

-Which bacteria are shown in PCRS and adenoid tissues? Answer: The bacteria involved in both of these areas have been shown to be very similar. Mostly S aureus, H influenza, and S pneumoniae. This has now been added in lines 96 – 100 with appropriate references. 

-What is maximal medical therapy for PCRS? Answer:There is no consensus on what necessarily constitutes maximal medical therapy and/or the duration based on guidelines. The options and preferred choices based on studies are discussed throughout the Treatment section starting on line 124.

-Which antibiotics are recommended by professional society guidelines for PCRS? Answer:The American Academy of Otolaryngology – Head and Neck Surgery guidelines on PCRS does not specify any specific antibiotics to use, but mention that culture-directed antibiotic therapy may improve outcomes for PCRS for those patients that have not responded to empiric antibiotic therapy. This is discussed on line 135 - 137

Reviewer 2 Report

This is an interesting manuscript successfully addressing the issue of the presence of adenoids in pediatric chronic rhinosinusitis. Authors should add a paragraph with recalcitrant cases, and  perhaps suggest different therapeutic approaches and/or considerations. Suggested ref: Severe Chronic Upper Airway Disease (SCUAD) in children. Definition issues and requirements. Karatzanis A, et al. Int J Pediatr Otorhinolaryngol. 2015 Jul;79(7):965-8.

Author Response

1)   This is an interesting manuscript successfully addressing the issue of the presence of adenoids in pediatric chronic rhinosinusitis. Authors should add a paragraph with recalcitrant cases, and  perhaps suggest different therapeutic approaches and/or considerations. Suggested ref: Severe Chronic Upper Airway Disease (SCUAD) in children. Definition issues and requirements. Karatzanis A, et al. Int J Pediatr Otorhinolaryngol. 2015 Jul;79(7):965-8. 

Answer: This is a good point. There has been a paragraph and discussion added on lines 177 – 185 with incorporation of suggested reference. 

Reviewer 3 Report

thank you for submitting this review

it would have been more valuable to perform a systematic review of literature on one aspect on PCRS to come up with new data and recommendations.

the present review is valuable for the general practitioner and pediatrician and not to the otolaryngologist 

Author Response

thank you for submitting this review

it would have been more valuable to perform a systematic review of literature on one aspect on PCRS to come up with new data and recommendations.

the present review is valuable for the general practitioner and pediatrician and not to the otolaryngologist 

-Answer: While this would be helpful, that is beyond the scope of what we were asked to write about. We were asked to write a review about the relationship of adenoids and pediatric CRS. We presented the newest data we could find, particularly new data since many of the guidelines have been written. 

Reviewer 4 Report

It is a well prepared paper

Author Response

It is a well prepared paper

--Answer: Thank you! We are very appreciative of that. 

Reviewer 5 Report

This review manuscript named “The Role of the Adenoids in Pediatric Chronic Rhinosinusitis” is a comprehensive review article with good logistic thinking and reasonable practical pathway. However, there are still some points should be concern.

In section 3, line 82, what is the meaning of CA? Please add the full name of CA.

In section 3, line 104, The “reduced susceptibility” should be amended to “increased susceptibility”.

Author Response

--In section 3, line 82, what is the meaning of CA? Please add the full name of CA. Answer:CA stands for chronic adenoiditis. I had previously referenced it prior to abbreviating it, but ended up deleting the sentence and had not spelled out chronic adenoiditis on line 82. It has now been corrected. Thank you! 

--In section 3, line 104, The “reduced susceptibility” should be amended to “increased susceptibility”. Answer:Thank you for noticing this. It has now been corrected.

Round  2

Reviewer 1 Report

Thank you for your kindly attention